# Recent Advances of In Vitro Culture for the Application of New Breeding Techniques in Citrus

**DOI:** 10.3390/plants9080938

**Published:** 2020-07-24

**Authors:** Lara Poles, Concetta Licciardello, Gaetano Distefano, Elisabetta Nicolosi, Alessandra Gentile, Stefano La Malfa

**Affiliations:** 1Food and Environment (Di3A), Department of Agriculture, University of Catania, Via Valdisavoia 5, 95123 Catania, Italy; lara.poles@phd.unict.it (L.P.); distefag@unict.it (G.D.); enicolo@unict.it (E.N.); stefano.lamalfa@unict.it (S.L.M.); 2CREA, Research Centre for Olive, Fruit and Citrus Crops, Corso Savoia 190, 95024 Acireale, Italy; concetta.licciardello@crea.gov.it; 3College of Horticulture and Landscape, Hunan Agricultural University, Changsha 410128, China

**Keywords:** regeneration, transformation, genome editing, genotype, agroinfiltration, promoter, selectable-marker genes, disease resistance

## Abstract

Citrus is one of the most important fruit crops in the world. This review will discuss the recent findings related to citrus transformation and regeneration protocols of juvenile and adult explants. Despite the many advances that have been made in the last years (including the use of inducible promoters and site-specific recombination systems), transformation efficiency, and regeneration potential still represent a bottleneck in the application of the new breeding techniques in commercial citrus varieties. The influence of genotype, explant type, and other factors affecting the regeneration and transformation of the most used citrus varieties will be described, as well as some examples of how these processes can be applied to improve fruit quality and resistance to various pathogens and pests, including the potential of using genome editing in citrus. The availability of efficient regeneration and transformation protocols, together with the availability of the source of resistance, is made even more important in light of the fast diffusion of emerging diseases, such as Huanglongbing (HLB), which is seriously challenging citriculture worldwide.

## 1. Introduction

Citrus is one of the most important fruit crops in the world. In 2018, the surface devoted to citrus production totalled 11.1 million hectares, with a huge production of oranges (75 million tons), followed by clementines, mandarins, tangerines and satsumas (34 million tons), lemons and limes (19 million tons), and grapefruits and pummelos (9 million tons) (FAOSTAT database results 2018 [1]).

However, the global citrus industry relies substantially on large-scale monoculture, and it is threatened by several diseases with a great economic impact in the main production areas, such as China, Brazil, Mexico, United States, and some Mediterranean countries.

The development of novel varieties with improved resistance to various pests and pathogens is one of the main aims of citrus breeding programs; conventional breeding strategy in citrus has demonstrated numerous limitations due to biological characteristics common to woody plants, such as long juvenile period, large size, long generation time, and also the lack of knowledge on how the most important horticultural traits are inherited. In addition, citrus display other limitations, such as nucellar polyembryony, self-incompatibility, and high heterozygosity, that genetic engineering and New Plant Breeding Techniques (NPBTs) [2,3] can overcome, leading to the development of novel varieties with the incorporation of selected traits, while retaining the unique characteristics of the original cultivar.

NPBTs include different biotechnological tools that are used to induce DNA modification, such as insertion, deletion, gene replacement, or stable gene silencing. Genome editing, or sequence-specific nuclease technology, involves the production of a permanent and inheritable mutation in a specific DNA sequence that can be inaccurately repaired by the plants’ own repair mechanism (leading to gene knock-out), or that can be accurately repaired using a DNA-repair template (leading to target mutation or gene replacement) [4,5,6].

Cisgenesis or intragenesis approaches are based on transformation with genetic material from closely related species capable of sexual hybridization, in contrast to transgenesis, where genetic material can be mixed between species; in particular, while cisgenesis involves the use of a copy of a complete natural gene, intragenesis allows in vitro recombination of different gene elements [7,8].

Other important techniques include trans-grafting, a method where a non-genetically modified (GM) scion is grafted on a GM rootstock leading to better performance of the top and to the production of GM-free fruits [3,8], and RNA interference (RNAi), a mechanism activated by the presence of target double-stranded DNA molecules that results in the inhibition or suppression of gene expression [9,10].

Compared to other fruit tree species, some citrus varieties are really amenable to tissue culture [11], and micropropagation and transformation have been widely used for many agronomically important varieties using different types of explants, such as epicotyls, shoot segments, protoplasts, and embryogenic cells.

Among the others, in vitro juvenile tissues are the most used due to their high morphogenic ability and the polyembryonic nature of many cultivars that enables the production of true-to-type plants by seed germination; however, this strategy cannot be adopted in seedless varieties, such as navel oranges and satsuma mandarins or monoembryonic species like clementine, where only the zygotic embryo develops from the seed [12]. To overcome this problem, other tissues with morphogenic potential, such as mature tissues or cell suspensions derived from embryogenic callus, need to be used for these cultivars [12,13,14].

This review aims to summarize the progress achieved in citrus genetic engineering, with particular focus on the transformation of juvenile and mature tissues, factors affecting the regeneration and selection of the transgenic shoots, and their main applications; new advances in citrus biotechnology, such as the use of selectable marker genes, inducible promoters, and genome editing will also be described. The availability of an optimized organogenesis protocol associated with an efficient *Agrobacterium*-mediated gene transfer system will contribute to a successful application of NPBTs and the development of novel varieties with improved quality features or resistance traits.

## 2. Regeneration of Citrus for Genetic Transformation

Citrus tissues are recalcitrant to regeneration and transformation; common systems use nucellar seedling internodes due to the polyembryonic nature of most citrus cultivars, but this cannot be applied to seedless genotypes or to species that are difficult to regenerate via organogenesis, such as mandarins. The use of cell suspensions or protoplasts obtained from embryogenic callus can represent a valid alternative for genetic transformation, to obtain plant recovery through somatic embryogenesis rather than the induction of adventitious shoots [12,15].

### 2.1. Genotype Influence

The availability of an organogenesis protocol based on the culture of juvenile explants allowed the production of transgenic plants for many citrus species, with variable degrees of success in terms of transformation efficiency (TE) [16,17]; genotype is one of the main factors influencing the effectiveness of the protocol, as some genotypes are considered easy to transform (e.g., citranges (*Citrus sinensis* L. Osbeck. × *Poncirus trifoliata* L. Raf.) [18,19], Duncan grapefruit (*C. paradisi* Macf.) [16,18]), while others are regarded as recalcitrant (e.g., Clementine (*C. clementina* Hort. ex Tanaka) [20] and sour orange (*C. aurantium* L.) [21]).

Despite the narrow genetic diversity present in citrus [22], the differences that exist between cultivars of the same species are sufficient to affect their organogenic response and the frequency of transformed explants after *Agrobacterium* infection (Table 1).

Among sweet orange cultivars, despite the fact that epicotyl explants of Valencia and Jincheng had similar regeneration potential (28.8% and 28.3%, respectively), they showed different percentages of TE, with the second cultivar being almost five-fold lower. The highest percentage is reached by *P. trifoliata*, a citrus related genotype with a short juvenile period particularly useful for functional genomics studies, and by Carrizo citrange, one of the most responsive species to transformation among citrus; in all cases, the transformation process reduces the percentage of regenerated shoots in all the examples shown (Table 1).

### 2.2. Source of Explant Type

Citrus epicotyls show a good in vitro morphogenic response, and therefore have been mostly used for the standardization of regeneration protocols [30]; considering mature shoots, internodes of 1 cm have been used for the regeneration and the transformation of adult tissues in citrus [20,31,32].

The use of thin sections of mature stems has been explored in sweet orange and has resulted in higher percentages of regenerated and transformed shoots (35% TE) [33,34] with respect to longer internodes.

Another alternative is the use of leaf discs, especially material from in vitro propagated or greenhouse-grown plants, which would assure abundant supply and a low risk of contamination. In sweet orange, only a few reports have been successful; no bud induction was obtained on leaf discs of mature Hamlin [31] or Thompson navel [35], while a regeneration rate of 60% was reached using Valencia [36], and the TE of its leaves was 23.33% [37].

Regeneration is possible through somatic embryogenesis that can be induced using appropriate culture media and starting from ovules of immature fruits [15]; the embryogenic cell suspension obtained can be maintained and transformed directly, via *Agrobacterium* infection, or indirectly, isolating protoplasts [38,39]. Citrus protoplasts obtained from leaves are not totipotent and do not develop into somatic embryos, while the ones obtained from embryogenic cell cultures have the best potential for proliferation and embryo regeneration [15]. Protoplasts are usually transformed using a polyethylene glycol (PEG)-mediated DNA uptake process or via electroporation [40,41].

### 2.3. Basal Media and Other Factors Influencing Organogenic Response

Basal culture media influence the morphogenesis performance, and the most stimulating are MS (Murashige and Skoog [42]) and MT (Murashige and Tucker [43]), irrespective of the cultivar analyzed [18,21,35,44,45,46,47]; Woody plant medium (WPM) [48] is mainly used for the elongation of adventitious shoots to ensure a larger dimension to facilitate micrografting in vitro [20,31,34].

The efficacy of the medium in the regeneration process, in terms of hormone concentration, has been investigated among different cultivars. Many reports have shown a promotive effect in citrus shoot regeneration using low cytokinin concentration (1–3 mg/L), depending on the cultivar [32,49].

The addition of cytokinin 6-Benzylaminopurine (BAP) was sufficient to induce organogenesis from mature and juvenile explants of many sweet orange genotypes, except for the Navelina cultivar; to increase the regeneration efficiency of this genotype, an auxine, 1-Naphthaleneacetic acid (NAA), was added to the regeneration medium containing BAP, increasing the percentages of callus growth, bud formation, and also TE (from 0 to 3%). The same treatment resulted in an opposite effect when applied to Pineapple genotype, and a reduction of TE from 6 to 0% was observed [49]. The same combination (BAP and NAA) in the regeneration medium gave good results for mature sweet oranges (Pera, Valencia, Natal, and Hamlin [31]), rangpur lime [44], and sour orange [21]; however, auxin supplementation did not improve the regeneration of Carrizo citrange, Mexican lime (*C. aurantifolia* Swingle), lemon, rough lemon, Cleopatra mandarin, *P. trifoliata*, *C. macrophylla*, and clementine [49,50].

When plant cells are subjected to stress, such as wounding or cutting, ethylene biosynthesis increases, affecting plant regeneration [51]. Different ethylene inhibitors have been evaluated in citrus tissue cultures. Among others, silver ions (Ag+) can interfere with ethylene receptors, improving cell regeneration; for example, the addition of AgNO_3_ had a weak effect on US-942 rootstock regeneration compared to the influence of other phytohormones [24].

In addition, the use of antioxidants can improve regeneration and TE; for example, lipoic acid improved the transformation of epicotyl segments of Mexican lime by five-fold compared to control explants [52].

Among the environmental conditions, photosynthetic radiation and incubation temperature are factors affecting the performance of in vitro tissue culture; in particular, it was reported that temperature of approximately 27 °C was adequate for the development of adventitious buds in sweet orange seedlings [53], and a period of incubation in darkness promotes an organogenesis response. Organogenesis from mature internodes of Pera, Valencia, Natal, and Hamlin oranges occurred directly from the explants without intermediate callus formation with a continuous 16-h photoperiod, and indirectly in darkness culture. Histological sections showed structural changes in the cambium with an intense cell proliferation at both cut ends after 15 days of culture (callus proliferation), and several meristematic regions differentiated from the callus tissues, leading to the formation of adventitious buds after 30 days [31]; similar observations were reported in sweet orange regeneration and grapefruit [24].

## 3. Citrus Transformation Protocols

Since the first attempt in citrus transformation in 1989 [54] that used a PEG-mediated strategy on protoplasts, *Agrobacterium*-mediated transformation has been shown to be the most widely used method, and approximately 90% of the transgenic plants were produced using this methodology [55].

The most used protocol for juvenile tissue explants transformation starts with the preparation of epicotyl segments; after *Agrobacterium* infection, explants are blotted dry and placed on cocultivation medium for 3 days at low light intensity. Subsequently, explants are transferred on regeneration and selection medium for 2 weeks in the dark until the formation of a white callus and then in a 16-h photoperiod with light; among the different protocols published for genetic transformation of citrus seedlings, few differences in the medium composition are reported [18,56,57]. Difficulties in the low rooting efficiency of regenerated shoots [19,58] were circumvented by the use of in vitro shoot micrografting [59] and minigrafting [60] onto decapitated seedlings of citrange germinated in vitro.

Besides many factors affecting the transformation, preincubation of explants in a hormone-rich medium prior to bacterial infection have been shown to increase the genetic transformation rates [16,18,25,61], activating cells at the cut end of explants and stimulating their divisions and de-differentiation. In juvenile tissues, 3 h of incubation in a MS medium supplemented with 13.2 μM BAP, 0.5 μM NAA and 4.5 μM 2,4-D was sufficient to increase the morphogenic competency in Carrizo citrange, Duncan grapefruit, Hamlin orange and Mexican lime [18], while for sour orange, a pre-culture of 1 day in MS medium containing 1 mg/L BAP and either 0.3 mg/L NAA or 0.3 mg/L 2,4-D, resulted in a stress response [21]; also in the transformation of stem segments of adult Tarocco oranges, the preincubation period of 6 h was sufficient to increase TE, while a prolonged period resulted in explant necrosis [25].

In mature tissues, a cocultivation phase after *Agrobacterium* infection in medium rich in auxin and in darkness conditions promotes hormone enrichment in the infected cells and stimulates callus formation [20,26,32,49].

The majority of *Citrus* species are recalcitrant to *Agrobacterium*-mediated transformation; in fact, this genus is not a natural host for *A. tumefaciens,* and so their mutual interaction has not evolved at the optimum level, as for other species [30]. To increase the rate of success, the disarmed hypervirulent *A. tumefaciens* strain EHA105, a derivative of the most virulent strain, A281 [26,62,63], or AGL-1 [56] were used, and the insertion of additional copies of *vir* genes from *A. tumefaciens* enhanced the transformation efficiency [20,63]. Acetosyringone, a phenolic compound secreted by wounded plant tissues, can stimulate *vir* gene activation, and its addition increased the TE in juvenile explants of Carrizo citrange [45] and sweet orange [18], but had no effect on Duncan grapefruit and Mexican lime [18].

For citrus species that are difficult to regenerate via organogenesis, such as mandarins, the use of cell suspensions or protoplasts obtained from embryogenic callus can represent a valid alternative for genetic transformation [12].

*Agrobacterium*-mediated transformation strategies using cell suspension cultures, seed-derived epicotyl segments, mature stem segments and PEG-mediated transformation using protoplast strategies were compared in the transformation of recalcitrant W Murcott (*C. reticulata* Blanco × *C. sinensis* L. Osbeck). Epicotyl segments and mature explants resulted in high regeneration efficiency (68% and 34%, respectively) and low TE (1.23% and 0.33%, respectively); juvenile cell suspensions and protoplasts showed higher TE, with values of 29% and 11%, respectively, with a large number of cells that were potentially amenable to transformation. Despite the fact that suspension cells offer the possibility to avoid chimeras due to the single-cell origin of regenerated somatic embryos, these techniques require a long time for regeneration and plant recovery compared to other strategies, and the regenerated plants are still juvenile, requiring years for production [12].

In addition, biolistic methods, recently applied for the transformation of epicotyl explants of Carrizo citrange with low TE (0.3–1.9 transgenic shoots per paired shot), can be optimized and become a valid alternative to *Agrobacterium*-mediated transformation [64,65].

Cultivation of plant cells and tissues with subsequent regeneration of the entire plants can be avoided using *in planta* transformation methods; this strategy was applied to Shatian pummelo (*C. maxima*), Jincheng and Xinhui oranges, leading to TEs of 20.41% [66], 46.3%, and 39.5%, respectively [67]; it is performed under non-sterile conditions and is faster than conventional tissue culture techniques; in fact, plants obtained using this method could be graft-propagated in 3 months post-transformation. Briefly, the apical meristem and primary leaves of pummelo seedlings were removed, and the decapitated epicotyls were winded by Parafilm to form funnels for *Agrobacterium* inoculation; then, funnels were removed, and wounds were wrapped with Parafilm and maintained in the dark. Following three days of co-culture, Parafilm wrap was removed, and cotton balls saturated with selection agent were used to soak the wounds of putative transformed seedlings three times. Seedlings were then wrapped again with Parafilm, kept in the dark for two weeks and then transferred to a greenhouse with natural lighting.

Citrus leaves can also be infiltrated with *Agrobacterium* for transient expression assays, useful for the characterization of gene function and the evaluation of candidate genes, e.g., Duncan grapefruit [68], Eureka Frost Nuclear [69], Eureka Frost, and Lisboa Frost lemons, and Troyer citrange [70].

Agroinfiltration procedure was implemented in Mexican lime using intermediate-aged leaves and setting *Agrobacterium* concentrations and buffer composition [71]. In addition, a pre-treatment with *Xanthomonas citri (Xcc)* before *Agrobacterium* infection significantly enhanced transient protein expression in different citrus species (Duncan grapefruit, Valencia orange, Key lime, Carrizo citrange, sour orange, and Meiwa kumquat), eliciting cell divisions [72,73]. *Xcc*-facilitated agroinfiltration was used to hasten transgene function assays in *Cre*/lox [73] and Cas9/sgRNA systems [73,74,75,76].

### 3.1. Selectable Marker Strategy

In most transformation systems, identification and selection of transgenic shoots are performed using genes that confer resistance to selective chemical agents, such as antibiotics or herbicides that are usually co-transformed with a gene of interest [16,56]; in citrus *nptII* (neomycin phosphotransferase II from *E. coli*), confers resistance to the antibiotic kanamycin, which is commonly used [77], but once the transformation has taken place, the marker gene is not useful anymore, and it represents an undesirable obstacle for biosafety issues and public concerns [27,78,79].

The *Citrus* genus is highly heterozygous, and its long generation cycles make the segregation and removal of marker transgenes in the progeny difficult [78,80].

Under non-selective conditions, transformed and non-transformed segments compete in the same space for shoot development, and non-transgenic events would be more competent to regenerate and prevailed over the transformed segments [49,78]; moreover, the selection of transgenic plants directly by molecular analyses could result in gene silencing [81,82] and in laborious, expensive, and time-consuming screenings. To avoid this risk, reporter markers, such as *β-glucoronidase (uidA* or *GUS*), which needs the extractive assay to be detected, and green fluorescent protein (*GFP*, Figure 1), a viable reporter gene, can be used to rapidly screen and select transformed shoots [20,81,83].

Type and concentration of the antibiotics influence the regeneration process, irrespective of the cultivar. De Oliveira et al. [44] evaluated the regeneration of 3 cultivars of oranges, Bahia, Valencia, and Pera, from adult tissues, testing different concentrations and types of antibiotics (timentine, cefotaxime, meropenem, and augmentin), and the best responses were obtained with 500 mg/L of cefotaxime. In addition, it was pointed out that, after transformation, in the selection and regeneration processes, different kanamycin concentrations had the smallest effects on the regeneration and TE compared to cytokinin type and concentration, and 50 mg/L kanamycin was sufficient to balance the growth of transgenic and non-transgenic cells [25].

Over the years, many efforts have been made to find alternative methods to replace the *nptII* selection system. One option is the phosphomannose isomerase (PMI)/mannose conditional positive selection system (*manA* gene [84,85]), which promotes the growth of transformed cells capable of synthesizing PMI enzyme on a medium that has mannose as a carbon source. It was first used by Boscariol et al. [23] in the transformation of sweet oranges, with TEs of 3–23% depending on the cultivar (Valencia 23.8%, Natal 12%, Pera 7.6%, and Hamlin 3%), and Ballestrer et al. [79] concluded that it was an excellent candidate for citrus transformation, yielding TEs of 30% for citrange epicotyls and 13% for sweet orange mature internodes. Recently, PMI selection has been applied for the biolistic transformation of Carrizo citrange to increase the TE obtained with kanamycin selection, 0.7%, to 1.9% transgenic shoots per shot, avoiding the introduction of antibiotic resistance in plants [64,65].

An ideal strategy for overcoming the biosafety problems associated with selectable marker genes is the direct production of transgenic plants containing only the gene of interest. Site-specific recombination systems enable the removal of the marker gene after the selection phase through the use of the multi-auto-transformation (MAT) vector system. This method combines two elements, first a positive selection using the isopentenyl transferase gene (*ipt*, which catalyzes the production of a precursor of several cytokinins [86]), and then a site-specific recombination system R/RS from *Zygosaccharomyces rouxii* [87], in which the R recombinase removes the DNA fragment placed between two recognition RS sites from the transgenic cells after transformation.

The MAT vector system was used in citrus transformation first by Ballester et al. [80], but the excision of the RS fragment was not always efficient and precise due to the constitutive expression of the R recombinase gene, and the *ipt* marker was clearly distinguishable in sweet orange [27], but not in citrange [80]. In 2008, Ballester et al. [79] improved the MAT vector system using an inducible R/RS-specific recombination system with transgenic-shoot selection through expression of the *ipt* gene and the indoleacetamide hydrolase/tryptophan monooxygenase (*iaaM*/H) marker gene, which causes the development of shoots exhibiting a characteristic shooty phenotype [88]. In this case, R recombinase gene expression was controlled by the inducible GST-II-27 promoter from maize [88,89], and the *uidA* reporter gene was included in the T-DNA but outside the RS fragment to facilitate the screening of regenerated shoots. The TEs obtained with this system were 7.2% for citrange and 6.7% in Pineapple orange, which were significantly lower if compared with kanamycin selection, which resulted in TEs of 40% and 15%, respectively; however, with this method, regeneration of non-transformed escape shoots was not precluded for any genotype [79].

Other site-specific recombination methods are based on the bacteriophage P1 *Cre*/loxP and on the yeast Flp/FRT [89]. In the *Cre*/loxP-mediated site-specific DNA recombination system, *Cre* recombinase specifically recognizes loxP sequences and performs a precise autoexcision of the DNA segment between the two sites [90]. This technology has been used by Zou et al. [27] in the genetic transformation of Jincheng orange. The vector includes an *ipt* gene and *Cre* recombinase inserted between the two loxP recognition sites, while the *GFP* reporter gene was located outside to monitor the transformation; both *Nosp* and Cauliflower mosaic virus (*CaMV35S)* promoters were evaluated in driving *Cre* recombinase expression, and the first was more suitable (100% deletion efficiency compared to 81.8% of *CaMV 35S*).

Problems of chimerism and inefficient deletions can be avoided by limiting the expression of the *Cre* gene with the use of tissue-specific [91,92,93,94,95] or inducible promoters, for example activated by heat shock [96,97,98].

### 3.2. Role of the Promoter

An important component to choose for the development of transgenic crops is the promoter element, which has an essential role in gene regulation at the transcriptional level; the characterization of gene regulatory sequences and their associated binding proteins provides valuable tools for plant genetic engineering.

A wide range of promoters derived from plants, viruses or bacteria has been used in plant genetic transformation. In *Citrus,* the most used is the *CaMV35S* promoter [99], which targets gene expression throughout the plant [18,21,45,46,47].

The availability of promoters and gene regulatory sequences derived from citrus is particularly important in the generation of intragenic or cisgenic plants, which use genetic material derived from the same species or from closely related ones. In addition, the availability of different constitutive promoters is important to avoid the risk of homology-dependent gene silencing caused by the use of the same constitutive promoters to express multiple transgenes [100]; Erpen et al. [101] identified the regulatory sequences from the *cyclophilin (CsCYP)*, *glyceraldehyde-3-phosphate dehydrogenase C2 (CsGAPC2*), and *elongation factor 1-alpha (CsEF1*) citrus constitutive genes, which exhibited constitutive gene expression in the vegetative tissues of transgenic Hamlin orange.

Additional studies on the regulatory elements of these promoters will enable the use of compact transformation vectors containing only the regulatory components instead of the entire plant promoter, considerably larger than the commonly used viral promoters [102].

In addition, in genetic engineering, a constitutive expression of the gene of interest is not always needed, and in many cases, gene expression could be limited to a particular developmental stage or particular organ or tissue. Promoters controlling spatio-temporal gene expression were evaluated in citrus. For example, the fruit-specific promoters that have been isolated thus far include the “type-3 metallothionein-like gene”, which confers preferential expression in juice sacs [103], and the *Cl111* promoter gene isolated from acid Eureka lemon and acidless lime (*C. limettioides* Tan.), which is pulp- and flower organ-specific [104]. For putative seed-specific expression, the *CuMFT1* promoter has been isolated from Satsuma mandarin (*C. unshiu* Marc.) [105].

Promoters that drive transgene expression preferentially to vascular systems were developed especially to target defence-related protein and to reduce or minimize expression in other parts of the plant. Among them, the citrus *phenylalanine ammonia-lyase (PAL) promoter (CsPP)*, which drives gene expression preferentially to xylem vessels, was useful against *Xylella fastidiosa* [106], while phloem-specific promoters could be useful for Huanglongbing disease, associated with a phloem-limited Gram-negative bacterium. Dutt et al. [107] evaluated the activity of four phloem-specific promoters in citrus transforming Mexican lime, and histochemical *GUS* analysis revealed vascular-specific expression of the gene at different levels, depending on the promoter. Rice tungro bacilliform virus promoter (RTBV [108]) was the most efficient, followed by *rolC* from *Agrobacterium rhizogenes* [109], then *Arabidopsis thaliana sucrose-H+ symporter* (*AtSUC2* [110]) and *Oryza sativa sucrose synthase l* (*RSs1* [111,112]).

Specific phloem gene expression was also studied in Hamlin and Valencia oranges using the promoters *C. sinensis phloem protein 2* (*CsPP2*), *A. thaliana phloem protein 2* (*AtPP2*), and *AtSUC2*; although the TE was low (from 0.2% to 4.5% among the two cultivars), the *attA* gene was preferentially expressed in the phloem [113].

Another possibility is the use of inducible promoters, especially pathogen-inducible promoters, to engineer plant lines with durable disease resistance and to avoid the presence and accumulation of antibacterial proteins in fruits.

The *A. thaliana heat shock protein 70B* promoter was used in an *Xcc*-facilitated agroinfiltration experiment for the temporal control of transgene expression in Duncan grapefruit; leaves subjected to agroinfiltration and incubation for 4 h at 42°C showed *GUS* staining, confirming the action of the inducible promoter in modulating *GUS* transient expression [73].

Zou et al. [114] evaluated the functionality of the pathogen-inducible promoters *PPP1* [115] and *hsr203J* [116] from tobacco and glutathione S-transferase (*gst1*) from potato [117] to drive expression of the *GUS* gene in response to the *Xanthomonas axonopodis* pv. *citri (Xac)* pathogen; the *PPP1* promoter was the most efficient promoter induced by *Xac* and wounding in transgenic Jincheng orange. The promoter *gst1* was used in 2009 by Barbosa-Mendes et al. [118] to drive expression of the *hrpN* gene (from *Erwinia amylovora* (Burr.)) in Hamlin transgenic plants and by Sendin et al. [119] to control the expression of the *Bs2* gene in Pineapple orange, both resulting in a reduced susceptibility to citrus canker.

The pathogenesis-related *PR5* gene promoter, which is rapidly induced after *X. citri* infiltration upon wounding [120], was used for driving the citrus *MAPK (CsMAPK1*) gene in Troyer citrange transgenic plants, reducing citrus canker symptoms [121].

Targeted expression is one of the most important aspects for the future development of value-added crops and for the application of NPBTs; public concerns about the use of pathogen-derived constitutive promoters have led to the isolation of plant-derived promoters that are more likely to be accepted and to the development of spatiotemporal gene expression that limits the presence of transgenes in the transformed cultivars [122].

## 4. Attempts to Reduce the Long Juvenility

Like other woody plant species, citrus has a long juvenile phase that prolongs the time for agronomic evaluation, delaying the release of new varieties; this characteristic becomes even more severe when the genetic improvement concerns rootstocks in which the level of polyembryony and the evaluation of the effect on scions require a very long time to be considered. For these reasons, the search for mature material to be used as explant source is of paramount importance, whereas most citrus genetic transformation systems utilize explants derived from juvenile tissues.

In citrus, the reproductive stage starts after 5 to 10 years; this period can be shortened by several biotechnological strategies, like the use of genes involved in flower initiation, the use of mature plant tissues or the use of genotypes with short juvenile periods, particularly useful for citrus functional genomics [28,123].

In the latter case, seedling stem segments of precocious trifoliate orange, an extremely early flowering mutant from *P. trifoliata* that has a juvenile period of 1–2 years, were used by Tong et al. [29] and by Tan et al. [124]; TEs of 57.4% and 20.7%, respectively, were recorded.

In addition, kumquat (*F. crassifolia* Swingle), a species close to *Citrus* that has a juvenile phase of only 2–3 years and bears fruit several times per year [28,125], and Mini-Citrus Hongkong kumquat (*F. hindsii*), which shows a very short juvenile period of approximately 8 months [126], were used.

The over-expression of flowering meristem identity genes in juvenile tissues leads to a shorter generation time and was first used in citrus by Pena et al. [127], transforming Carrizo citrange seedlings with the *Arabidopsis LEAFY (LFY*) or *APETALA1 (AP1*) genes and reducing the juvenility phase of transformed plants from 7 years to 12–20 months. In the transgenic plants obtained, flowering remained under endogenous and environmental controls, and the new feature was inherited by the offspring; in particular, AP1-transgenic citranges were fully normal and behaved as rapid-cycling trees, showing a generation time of approximately one year from seed to seed, allowing faster propagation and genetic transformation studies, making possible the rapid evaluation of flower or fruit traits [128]. The strategy was also applied to Meiwa kumquat, which showed a TE of 4.08% using epicotyl segments [123].

Generation time was also reduced by the constitutive expression of the *CiFT* gene in trifoliate orange, the citrus homolog of the flowering-time (*FT*) gene in *Arabidopsis*; transformants flowered 12 weeks after being transferred to the greenhouse [129,130]. Pineapple orange was transformed to increase their β-carotene content with the simultaneous overexpression of the FLOWERING LOCUS T from sweet orange (*CsFT*); early fruiting phenotype (approximately 1 year after being grafted in the greenhouse) was observed, with two fruiting cycles per year displayed by transgenic plants, which enabled a rapid characterization of fruit quality traits [131].

Another strategy is the use of virus vector based on citrus leaf spot virus (CLBV) expressing the *CiFT* gene, which promotes precocious flowering within 4 to 6 months in juvenile plants of several citrus species [132].

Finally, the genetic transformation from mature tissues represents a valid strategy to bypass the long juvenile phases and to decrease the time and cost for the obtainment of new varieties for which fruit characteristics must be evaluated for years. For these reasons, quick and easy protocols for transformation of mature tissues are required to accelerate functional genomics studies, including a better understanding of genes underlying quality traits [49].

The use of adult tissues in fruit crops for in vitro culture is hampered by the high level of contamination and the reduction or loss of morphogenetic abilities [31], in fact, the transition between juvenile and adult stages results in a progressive loss of competence for organogenesis and embryogenesis [133]. However regeneration from adult somatic tissues is highly recommended for clonally vegetatively propagated fruit tree crops, in order to maintain genetic uniformity of the cloned plants, especially for the highly heterozygotic species, such as citrus.

Transformation of mature tissue of citrus was first described by Cervera et al. [26] and has proven to be successful in the transformation of Pineapple orange [26], where it has led to the production of transgenic plants (6.1% TE) flowering and bearing fruits in 14 months after being transferred to the greenhouse; this system is also a valid alternative for the transformation of citrus seedless and monoembryonic varieties and was patented in Europe and the United States [62]. The protocol included three steps, starting with the *ex vitro* invigoration of source plant material by grafting adult buds onto juvenile vigorous rootstock, such as *C. volkameriana*. A second step consists of the optimization of tissue culture conditions to shift explant citrus cells to a competent state for *Agrobacterium*-mediated transformation and regeneration; explants are usually sterilized, co-cultivated with the engineered *A. tumefaciens* for 15 min, blotted dry on sterile paper and placed horizontally on co-cultivation medium rich in auxins for three days, with cocultivation at low light intensity. Internodes are then transferred to regeneration medium containing opportune hormones and antibiotics and are maintained in the dark for 2–4 weeks. Finally, in the third step, regenerated shoots are checked for their transgenic nature, micrografted onto decapitated seedlings of Troyer citrange germinated in vitro and acclimatized.

This method was also optimized for the transformation of the more recalcitrant clementine increasing transgenic plant regeneration efficiency of this genotype from 0.3 to 3% [20], although this genotype showed lack of bud uniformity in sprouting and morphology. In particular, the concentration of 2,4-D used in the co-cultivation medium was doubled from 2 to 4 mg/L (increasing transformation frequency by 1.7- to 2.3-fold), the co-cultivation period was reduced from three to two days, and the dark period after co-cultivation was extended from 2–4 to 5–6 weeks. As clementine was more recalcitrant to *A. tumefaciens* infection, additional copies of *virG* and *virE* were introduced into the plasmid used in the transformation, which led to a consistent enhancement of transformed plants obtained from 1.5- to 2.3-fold. In clementine, only regeneration under non-selective conditions was adequate to recover a sufficiently large number of transgenic shoots, distinguished by the *GUS* test or *GFP* visualization; they also lowered the kanamycin concentration to 25 mg/L. However, while an increase in callus induction was observed, shoot regeneration remained low. The same authors report that using WPM medium instead of MS, longer shoots, easier to be micrografted, were obtained (4 mm instead of 2 mm length) [20].

He et al. [134] used a novel *Agrobacterium*-mediated transformation system for mature auxillary buds leading to TEs of 7.5% for Jincheng and 8.3% for Newhall, both commercial orange cultivars. This method involved the use of mature shoots and did not contemplate the use of either hormones, antibiotics selection and solid medium, because all passages were carried out on MS liquid medium and a filter paper bridge.

Transgenic plants obtained start to blossom and bear fruits in the second year after the last grafting. The high-frequency transformation was attributed to the use of rootstock that enhanced nutrition for shoot development, to the absence of kanamycin selection and to the regeneration ability of the auxillary meristem in micrografted shoot.

An optimized protocol for mature tissue transformation was published in 2015 by Orbović et al. [56], with a TE of 12.8% using Hamlin orange; compared to the previous protocols, this included a stronger sterilization process and the addition of another antibiotic selection (10 mg/L of Meropenem during the first 2 weeks of selection).

Adult tissues (stem segments) of Tarocco orange were transformed with a TE of 11.7% [25] using a protocol that included a pre-incubation step, commonly used for the transformation of juvenile material but never employed in mature explant transformation experiments.

The highest percentage for the transformation of mature tissue was reached with Pera orange, with 35% of transgenic plantlets [33] using thin transversal segments (1–2 mm) of newly elongated shoots from greenhouse plants instead of internodal segments.

## 5. Success in Transgenesis Applied to Citrus

Genetic engineering has been strongly considered for the development of novel citrus varieties, offering a wide range of tools and strategies that enable the insertion or the editing of desirable traits into elite commercial cultivars. The applications of transgenesis are wide and include resistance to biotic and abiotic stresses and the control of fruit quality traits.

Several traits have been considered for genetic transformation, including early flowering (See Section 4, ‘Attempts to reduce the long juvenility’), tree architecture and growth habitus [135,136,137], tolerance to abiotic stresses [138,139,140,141], improvement of fruit quality [41,142,143], in particular carotenoid content [131,144], and seedlessness [13,145,146].

Thus far, the main aspects rely on biotic stresses, as these are the most limiting factors for citriculture worldwide. In the last years, great interest has been devoted to the development of novel varieties showing resistance to citrus greening (Huanglongbing, HLB).

HLB is considered the most devastating citrus disease worldwide (FAO2015); for example, the citrus utilized production in the United States in the 2017–2018 season (6.13 million tons) was decreased by 20% from the 2016–2017 season and by 66% with respect to the record high production of the 1997–1998 season (17.8 million tons); moreover, Florida’s on-tree value of the 2017–2018 citrus crop ($551 million) was the lowest since the 1976–1977 season ($530 million) [147]. Greening is associated with 3 species of phloem-restricted Gram-negative bacteria: *Candidatus Liberibacter asiaticus* (CLas) and *C. Liberibacter americanus*, which are transmitted by the Asian citrus psyllid *Diaphorina citri*, and *C. Liberibacter africanus*, which is transmitted by the insect *Trioza erytreae* [148,149,150]. No curative methods are available for the disease; to ensure citrus survival in Europe, preventive measures are currently being developed within an European project (www.prehlb.eu).

Different strategies can be adopted to confer disease resistance to citrus cultivars. Genetic transformation with the constitutive expression of antimicrobial peptides (AMPs), a set of peptides of the innate immunity with antimicrobial activity [151], has been used to control bacterial diseases, such as HLB and citrus canker [152]. In citrus the most used AMPs are insect-derived *attacin A* [46,47,113,153], *creopin B* and *Shiva A* [134,154], *sarcotoxin IA* [33], a *thionin* [155] and *dermaseptin* [156].

In addition, the introduction in plants of resistance genes (R-genes) coding for proteins that recognize pathogen avirulence gene products (avr-gene [157]) can confer race-specific resistance, e.g., the pepper R-gene *Bs2* used against citrus canker [69,119].

Another possibility is the use of heterologous expression of receptors, which identify conserved molecules in the pathogen and trigger the plant’s immune response to a wide range of microorganisms; to enhance citrus canker resistance, the genes that have been considered are *Xa21*, a receptor kinase-like protein from rice [158,159], and the *Flagellin Sensitive 2 (FLS2)* receptor gene from *Nicotiana benthamiana* [160].

Alternative approaches against pathogen diseases have aimed to enhance the systemic acquired resistance (SAR), the plant’s inducible defence mechanism that increases innate resistance to further infection by pathogens [161]. The SAR response is induced by salicylic acid and is associated with the production of pathogenesis-related (PR) proteins that confer long-lasting broad-spectrum resistance; in citrus, this strategy was used against citrus canker [162,163,164] and HLB using *NPR1* [165,166,167].

To improve plant defence against fungi, the overexpression of genes encoding products with in vitro antifungal activity has been used, e.g., the *chit42* gene from *Trichoderma harzianum,* leading to an increased resistance of transgenic lemons to different fungi (such as *Phoma tracheiphila* and *Botrytis cinerea* [168,169] and conferring resistance to some post-harvest pathogens [170].

Pathogen-derived resistance was used against Citrus Tristeza Virus (CTV), which replicates in phloem vessels and is transmitted by *Toxoptera citricida*, an aphid vector; the *p25* coat protein from CTV was used to transform Mexican lime [171], and it was demonstrated that plants exhibiting post-transcriptional gene silencing (PTGS) also showed resistance to CTV due to the accumulation of p23-specific small interfering RNAs (siRNAs) [172]. RNAi, the approach that involves the knockdown of gene expression mediated by siRNAs using specific double-stranded RNA molecules, was applied to control CTV [173,174], citrus psorosis virus [175], citrus canker [70] and fungal pathogens, such as *Alternaria alternata* [176] and *Phytophtora* spp. [177].

Commercialization of disease-resistant citrus cultivars will presumably take many years, but the development of resistant or tolerant new genotypes that will replace susceptible varieties is one of the most realistic long-term solutions to many devastating diseases, such as HLB; until that time, it is important to incentivize cooperation in pest and disease management to guarantee vector control and tree monitoring [178,179].

## 6. Genome Editing

One of the most important NPBTs is genome editing, a technology based on programmable nucleases that produce site-specific DNA double-strand breaks (DSBs), which trigger endogenous DNA repair systems, resulting in targeted modification. The first tool used was zinc-finger nuclease (ZFN) followed by transcription activator-like effector nucleases (TALENs) in 2011. Since 2013, clustered regularly interspaced short palindromic repeats (CRISPR)-associated (Cas) nucleases have become the most popular method for plant genome editing [180].

In the CRISPR-Cas system, an adaptive immune system of prokaryotes [181], Cas nuclease is directed by a single guide RNA (sgRNA) that recognizes a target DNA sequence flanked by a protospacer adjacent motif (PAM) and generates specific DSBs. Nuclease-induced DSBs can be repaired by the non-homologous end-joining (NHEJ) pathway, which leads to the introduction of insertion/deletion mutations (*indels*) of various lengths, or by homology-directed repair (HDR), which is useful to introduce specific point mutations or to insert desired sequences through recombination of the target locus using DNA ‘donor templates’ present at the moment of DSB formation [182].

Since the first application of genome editing in plants, much progress has been made in the development of CRISPR-based editing tools; numerous Cas variants and orthologs with specific PAMs have been discovered together with precise genome editing by base editors, expression systems for multiplexing, transcription regulation and epigenome editing [183].

In citrus, Jia and Wang [72] reported the first genome editing using the Cas9/sgRNA system and *Xcc*-facilitated agroinfiltration on Valencia orange. The delivery of Cas9 and sgRNA were accomplished with a particular agroinfiltration that consists of an initial inoculation of *Xcc* followed by an *Agrobacterium* infiltration on Valencia leaves; the target gene was the endogenous *Citrus* phytoene desaturase (*CsPDS*) gene, an enzyme required for the biosynthesis of carotenoid pigments that results in a white-colored (albino) phenotype when it is silenced or mutated [184]. The mutation rate was approximately 3.2–3.9%, with no off-target mutagenesis detected. Jia and Wang [73] applied the same strategy on Duncan grapefruit and, being a grapefruit hybrid between pummelo and sweet orange [185], they were able to apply the Cas9/sgRNA system to specifically modify one of the two *CsPDS* alleles of the variety.

Subsequent application of genome editing has focused on editing genes involved in citrus disease resistance, especially in citrus canker.

Most of the studies were performed to target the *CsLOB1* gene (*C. sinensis Lateral Organ Boundaries 1*), a disease-susceptibility gene upregulated by *PthA4*, a transcription activator-like effector of *Xcc* [186,187], in particular to target the effector binding elements (EBEs) of PthA4, which are located in the promoter of the *CsLOB1* gene (*EBEPthA4-CsLOBP*), and should confer resistance to the disease without losing *CsLOB1* function.

Peng et al. [188] edited Wanjincheng orange using 5 different constructs to modify different regions along *EBEPthA4-CsLOBP*; through the transformation of epicotyl segments, they obtained 16 lines (42% TE) with *EBEPthA4* modifications and 4 mutation lines that showed enhanced resistance to citrus canker.

Duncan grapefruit epicotyl transformation was achieved by Jia et al. [74] and resulted in 4 lines with targeted modification of only *EBEPthA4 CsLOBP Type I* with a mutation rate of 15.63–81.25%; the transgenic plants were susceptible to *Xcc* infection. In 2017, Jia et al. [75] succeeded in disrupting the coding regions of both alleles of *CsLOB1,* and no canker symptoms were observed in the lines DLOB9 (mutation rate of 89.36%), DLOB10 (88.79%), DLOB11 (46.91%), and DLOB12 (51.12%) after *Xcc* inoculation. In both studies, no off-target mutation was detected, but only a few among the possible off-targets were subjected to analysis; an alternative strategy to reduce off-target mutations is the use of a different type of nuclease, such as CRISPR derived from *Prevotella* and *Francisella* (CRISPR-Cpf1), a new class II CRISPR-Cas system [189,190] that has been used to edit tobacco, rice and soybean [191,192,193,194]. In comparison with Cas9, Cpf1 exhibits little to no off-target activities in plant cells [195], has a different protospacer adjacent motif (T-rich PAM instead of G-rich one, NGG), generates cohesive ends with four or five nucleotide overhangs (compared with SpCas9, which produces blunt ends), promoting an HDR mechanism, and among the other features, Cpf1 requires shorter CRISPR RNAs (crRNAs 43 nucleotides instead of 100 of Cas9), making this system more suitable for multiplexed genome editing [189,190]. *Lachnospiraceae bacterium ND2006* Cas12 (LbCas12a) was used to edit Duncan grapefruit *EBEPthA4-CsLOBP*; epicotyls were transformed via *Agrobacterium*, and the biallelic mutation efficiency obtained was 5%, with no off-targets observed [76].

Recently, Jia et al. [196] published a protocol for the application of the CRISPR/Cas system via *Agrobacterium*-mediated transformation of epicotyl tissues in citrus, and the CRISPR/Cas9 system has been applied to Mini citrus Hongkong kumquat. Despite the low TE of *Agrobacterium*-mediated transformation (0.2–4%), it was possible to apply CRISPR/Cas9 and achieve a T1 generation in approximately 15 months; the modifications of target genes in the CRISPR-modified *F. hindsii* were predominantly 1-bp insertions or small deletions, and all T1 seedlings showed a mutation rate of 100% at the sgRNA1 targeting site.

Another approach was used by Wang et al. [197], editing the transcription factor *CsWRKY22* that was negatively correlated with citrus canker resistance. Epicotyls of Wanjincheng orange were transformed, and the transgenic plants W1-1, W2-2, and W2-3 showed 85.7%, 79.2%, and 68.2% mutation rates, respectively, with off-target frequencies of 3.0-16.0%; resistance evaluation indicated that transgenic plants delayed the development of canker symptoms.

Although all these studies demonstrate how CRISPR/Cas9 technology can be exploited for citrus genome editing, accelerating the breeding process and combining multiple favorable traits, there is a need for more precise biotechnology tools than those that are currently available.

One of the problems is the efficiency of the editing obtained; despite the fact that several computational tools are now available for designing guide RNAs targeting a specific gene, the editing efficiencies might be different due to the existence of variant alleles not included in online citrus genome databases [185]; for this reason, the investigation of the sequence of the gene of interest [74,75,188], the functionality evaluation of many sgRNAs using *Xcc*-facilitated agroinfiltration [72,74,75,76], and the in vitro cleavage analysis of the construct before citrus transformation [197] represent fundamental steps to increase editing efficiency.

The low frequencies of mutations induced by the CRISPR/Cas9 system used in citrus were improved by Zhang et al. [198], who used a different promoter to drive Cas9 expression, replacing the *CaMV35S* promoter with the *A. thaliana YAO* sequence [199] and increasing the frequency of mutational events from 3.2–3.9% [198] to 75% using the same sgRNA. Le Blanc et al. [200] also demonstrated that temperature has an effect on mutation rate achieved by the CRISPR/Cas9 system; Carrizo citrange transgenic plants containing *pYAO:SpCas9* and sgRNA targeting *CsPDS* genes that were exposed to several heat stress treatments (24 h at 37 °C and 24 h at 24 °C repeated seven times) showed an increase in targeted mutagenesis (100% *CsPDS* alleles mutated) with respect to those continuously grown at 24 °C (approximately half of the *CsPDS* alleles mutated). This result suggests that all CRISPR/Cas9 systems require higher temperatures to achieve optimal editing efficiency, regardless of the promoter used to regulate Cas9 expression [200], and that many aspects of the functioning of this technology are still to be explored.

Jia and Wang [201] generated homozygous and biallelic canker-resistant pummelo in the T0 generation via the CRISPR-Cas9 system with a 100% mutation rate in the *EBE* region of the *LOB1* promoter. Zhang et al. [198] also developed a bifunctional selectable and visible marker for citrus (*eGFP-NPTII*) that improved the recovery of transgenic events expressing high levels of Cas9, reducing the number of promoters present in the vector. In citrus, special efforts to control CRISPR/Cas9-mediated chimeric mutation are required, and the optimization of regeneration protocols will offer a great opportunity to select transgenic events and reduce the formation of chimeric mutations [197].

Other options include the use of embryogenic calli transformation that rarely produce transgenic chimeras [12,13] and a transient approach using purified CRISPR/Cas9 ribonucleoproteins to edit plant protoplasts, which has been tested in wheat [202] and applied to grape and apple [203].

Other concerns are related to the findings of new target genes for editing and to genetically modified organisms legislation; knowledge of plant pathogen interactions and mechanisms is critical to the development of new varieties with improved quality or resistance to disease via the CRISPR/Cas system [204]. The legislation of genome-edited plants is still a debated issue at international scientific and political forums, and many countries are in the process of drafting the regulatory frameworks for their use [205].

## 7. Conclusions

The development of novel citrus varieties with improved quality and resistance to biotic and abiotic stresses is one of the main purposes of breeding programs. Thus far, the use of conventional breeding techniques in citrus has been shown to be time consuming and difficult due to the many limitations of typical of tree crops, such as the long juvenility and high heterozygosity.

The application of NPBTs could overcome these problems, offering new tools that combine site-specific and targeted editing with a reduction in the time for plant breeding, thus leading to lower production costs. Many aspects need to be considered to apply transgenesis to citrus, among them: (i) the organogenic response is largely genotype-dependent, and (ii) the regeneration efficiency for many commercial varieties is still low. Other aspects of great relevance rely on the establishment of appropriate strategies to limit the expression of the transgenic gene in a particular organ and on techniques to efficiently remove selectable marker gene(s).

Despite the numerous papers published over the last several years, the availability of new sequencing data has greatly advanced the knowledge on genes underlying pathways of interest. This aspect will certainly offer new opportunities for the establishment of targeted breeding programs. The availability of germplasm collections encompassing a high fraction of the allelic variability characterizing Citrus heirloom varieties and/or landraces represents a valuable genetic reservoir that can be readily transferred into other varieties through NPBTs for the definition of novel varieties characterized by superior agronomical traits.

## Figures and Tables

**Figure 1 plants-09-00938-f001:**
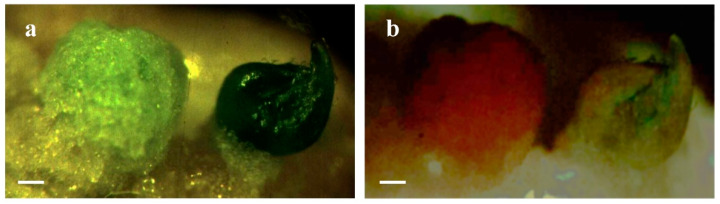
Callus and shoot in Troyer citrange internode observed under a stereomicroscope with white light (**a**) and 480 nm-excited blue light (**b**); in the latter case, it is possible to discriminate the ‘escape’ callus (red) and the fluorescent transgenic shoot (green). Bar = 1 mm.

**Table 1 plants-09-00938-t001:** Regeneration (RE) and transformation (TE) efficiencies of different citrus species. Explant types considered are epicotyl segment (ES), mature internode segment (MIS), and mature nodal segment (MNS) with buds removed.

Species	Cultivar/Genotype	Explant Type	RE (%)	TE (%)	Reference
*C. sinensis* L. Osb. × *P. trifoliata* L. Raf.	Carrizo citrange	ES	37.5	20.6	[19]
*C. sinensis* L. Osb.	Valencia	ES	28.8	23.8	[23]
*C. sinensis* L. Osb.	Valencia	MIS	9.12	0.88	[24]
*C. sinensis* L. Osb.	Tarocco	MNS	74.7	9.1	[25]
*C. sinensis* L. Osb.	Pineapple	MIS	23	6.1	[26]
*C. sinensis* L. Osb.	Jincheng	ES	28.3	4.7	[27]
*Fortunella crassifolia* Swingle	Jindan	ES	13	3.6	[28]
*C. reticulata* ‘Sunki’ × *P. trifoliata* ‘Flying Dragon’	US-942	MIS	29.42	3.96	[24]
*C. clementina*	Clemenules	MIS	1.28	0.3-3	[20]
*C. paradisi* Macf.	Ruby Red	MIS	10.70	1.05	[24]
*C. medica* L.	Etrog	MIS	9.49	1.49	[24]
*P. trifoliata* L. Raf.	Precocious trifoliate orange	ES	66.1	57.4	[29]

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
