# Peer review of "Recent Advances of In Vitro Culture for the Application of New Breeding Techniques in Citrus"

_plants, 2020, doi:10.3390/plants9080938_

Round 1
Reviewer 1 Report
This is an interesting, well-designed, and well-written review on the recent advances of in vitro regeneration and transformation procedures which can be used for the genetic improvement of citrus, with a specific focus on New plant breeding techniques (i.e. RNAi, gene editing). However, I recommend some minor revisions before the manuscript is considered for publication.
- Please check carefully the reference numbers in the whole text, according to Plants format references must be numbered in order of appearance in the text. For example, at line 41 references should be numbered as (1, 2) and not (1, 189), the same for table 1, etc...
- There are some paragraphs, especially in the introductive chapter, which need to be associated to appropriate references, in particular I suggest to include some more references:
- From line 45 up to line 49
- From line 50 up to line 53
- From line 54 up to line 58
- From line 87 up to line 89
- From line 89 up to line 92
- From line 119 up to line 122
- From line 144 up to line 145
- From line 234 up to line 239
- From line 366 up to line 369
- Please check reference n. 36 at line 172, it seems that in this patent no procedures based on the use of citrus juvenile explants for the genetic transformation trials are reported.
- I suggest to add some more appropriate references for lines 176-177.
- Lines 179 and 182: please describe better which kind of incubations were done in these experiments.
- Line 122: please check the sentence “…kept in the dark two FOR weeks…”
- Figure 1: this figure has a very low resolution, I suggest to use the same image with a higher resolution.
Line 259: please correct "process" with "processes".
Line 278: please close the brackets.
Line 304 and 389: please write “Cre” in italics, the same for FT gene.
Line 412: I don’t agree with the authors. Regeneration from adult somatic tissues is highly recommended for clonally propagated crops, such as fruit tree species, in order to maintain genetic uniformity of the cloned plants, especially for highly heterozygotic species.
Line 493: please delate “oooor” at the end of this sentence.
Line 514: please check the names of all plant viruses cited in the manuscript and write them following a unique format, if possible according to the ICTV rules (no italics, no capital letters to my knowledge).
Author Response
Report 1
This is an interesting, well-designed, and well-written review on the recent advances of in vitro regeneration and transformation procedures which can be used for the genetic improvement of citrus, with a specific focus on New plant breeding techniques (i.e. RNAi, gene editing). However, I recommend some minor revisions before the manuscript is considered for publication.
Dear reviewer, thanks for your word of appreciation.
- Please check carefully the reference numbers in the whole text, according to Plants format references must be numbered in order of appearance in the text. For example, at line 41 references should be numbered as (1, 2) and not (1, 189), the same for table 1, etc...
Thanks for noticing it, the references were carefully checked and renumbered when needed.
There are some paragraphs, especially in the introductive chapter, which need to be associated to appropriate references, in particular I suggest to include some more references:
- From line 45 up to line 49
- From line 50 up to line 53
- From line 54 up to line 58
- From line 87 up to line 89
- From line 89 up to line 92
- From line 119 up to line 122
- From line 144 up to line 145
- From line 234 up to line 239
- From line 366 up to line 369
Thanks for your suggestion, references were included to support the statements in the above mentioned lines.
- Please check reference n. 36 at line 172, it seems that in this patent no procedures based on the use of citrus juvenile explants for the genetic transformation trials are reported.
Right, this reference was removed from the new version of the paper
- I suggest to add some more appropriate references for lines 176-177.
Thanks for the suggestion, 3 more references were included
- Lines 179 and 182: please describe better which kind of incubations were done in these experiments.
Thanks for the suggestion, this part was better described in the new version of the manuscript
- Line 122: please check the sentence “…kept in the dark two FOR weeks…”
Right, we guess reviewer referred to line 222, we made the proposed change
- Figure 1: this figure has a very low resolution, I suggest to use the same image with a higher resolution.
Thanks for the comment, we included a new version of the figure with better resolution (300 dpi)
Line 259: please correct "process" with "processes".
Thanks for noticing it, we changed the text accordingly
Line 278: please close the brackets.
Thanks for noticing it, we changed the text accordingly
Line 304 and 389: please write “Cre” in italics, the same for FT gene.
Thanks for noticing it, we changed the text accordingly
Line 412: I don’t agree with the authors. Regeneration from adult somatic tissues is highly recommended for clonally propagated crops, such as fruit tree species, in order to maintain genetic uniformity of the cloned plants, especially for highly heterozygotic species.
We agree with the reviewer, we changed the text following your suggestion
Line 493: please delate “oooor” at the end of this sentence.
Thanks for noticing it, we corrected the error
Line 514: please check the names of all plant viruses cited in the manuscript and write them following a unique format, if possible according to the ICTV rules (no italics, no capital letters to my knowledge).
Thanks for noticing it, we changed the text according to your suggestion
Reviewer 2 Report
The work entitled "Recent advances of in vitro culture for the application of new breeding techniques in citrus" by Poles et al., is an excellent work that supports the aim of the review paper. The presented data will facilitate future efforts towards the development of new citrus varieties via the implementation of New Plant Breeding Techniques (NPBTs).
The overall text is very well written, very comprehensive, and easy to follow. All the presented scientific data are very well presented and correspond to the most recent advances in science. It is an excellent work that will meet the criteria of the current journal and contributes to the development of novel breeding techniques in citrus.
The presented paper could be published in its current form.
Author Response
Report 2
The work entitled "Recent advances of in vitro culture for the application of new breeding techniques in citrus" by Poles et al., is an excellent work that supports the aim of the review paper. The presented data will facilitate future efforts towards the development of new citrus varieties via the implementation of New Plant Breeding Techniques (NPBTs).
The overall text is very well written, very comprehensive, and easy to follow. All the presented scientific data are very well presented and correspond to the most recent advances in science. It is an excellent work that will meet the criteria of the current journal and contributes to the development of novel breeding techniques in citrus.
The presented paper could be published in its current form.
Dear reviewer, your words of appreciation reward the efforts we made in writing a review that could be of interest for researchers working on this field. We are glad you find our manuscript publishable in its current form.